# Multifunctional Silver Nanowire Fabric Reinforced by Hot Pressing for Electromagnetic Interference Shielding, Electric Heating, and Sensing

**DOI:** 10.3390/polym15214258

**Published:** 2023-10-30

**Authors:** Fangmeng Zeng, Yiqian Zheng, Yuxin Wei, Han Li, Qicai Wang, Jian Shi, Yong Wang, Xinghua Hong

**Affiliations:** 1International Silk Institute, Key Laboratory of Intelligent Textile and Flexible Interconnection of Zhejiang Province, College of Textiles, Zhejiang Sci-Tech University, Hangzhou 310018, China; zfmeng@zstu.edu.cn (F.Z.); yiqianz2023@163.com (Y.Z.); 2023210201059@mails.zstu.edu.cn (Y.W.); lihan@zstu.edu.cn (H.L.); qcwang@zstu.edu.cn (Q.W.); 2Hubei Key Laboratory of Low Dimensional Optoelectronic Materials and Devices, Hubei University of Arts and Science, Xiangyang 441053, China; 3Department of Materials Science and Engineering, National University of Singapore, Singapore 117575, Singapore; 4School of Textile and Garment, Anhui Polytechnic University, Wuhu 241000, China

**Keywords:** silver nanowires, fabric, hot pressing, durability, multifunction

## Abstract

Durability and multifunctionality are crucial considerations in the realm of electronic textiles. Herein, a hot-pressing process has been developed to enhance the fixation of silver nanowires (AgNWs) on polyethylene terephthalate (PET) fabric. The conductivity, electromagnetic shielding, and electric heating properties of the hot-pressed fabric were measured to demonstrate the effectiveness of the hot-pressing process. The conductivity of the hot-pressed fabric (180 °C for 90 s) was found to be 464.2 S/m, while that of the fabric without hot pressing was 94.9 S/m. The washed hot-pressed fabric was able to provide a maximum electromagnetic shielding of 17 dB, a negative strain sensing performance (the ΔR/R_0_ of the hot-pressed fabric was maintained at −15%), and an outstanding electric heating property (the temperature reached 110 °C at a current of 0.08 A). This AgNW fabric holds great potential for use in multi-functional wearable devices, and the hot-pressing process improved its stability and durability, making it suitable for industrial production.

## 1. Introduction

The rapid advancement of artificial intelligence has led to an increased interest in wearable electronic fabrics that can be used for monitoring health, thermal regulation, and shielding protection. The development of multifunctional flexible fabrics with potential applications in electromagnetic shielding, wearable sensors, healthcare monitoring, and human–machine interactions is imperative to meet the growing demand [1,2]. For instance, research has been conducted on conductive cotton fabrics that exhibit dual functionalities, such as electrical heating and electromagnetic interference shielding [3], in addition to capabilities such as photothermal conversion and sensing ability [4]. Such research highlights the importance of developing advanced wearable fabrics with versatile functionalities to address the diverse needs of modern society [5].

In addition to possessing multiple functionalities, a multifunctional flexible fabric should exhibit adequate stability and durability. Among all fabrics, polymeric fabrics are widely regarded as the optimal substrate for flexible wearable sensors due to their inherent advantages such as comfort, flexibility, and easy manufacturing [6]. PET, a commonly used commercial synthetic fiber, boasts high tenacity and strength, washability, wearability, dimensional stability, and resistance to various chemicals [7,8]. However, the quality of smart PET fabrics primarily hinges on the electrical conductivity of the fabric. The electrical conductivity of the fabric is a critical factor for the quality of smart fabrics.

Metal nanowires with high length/diameter ratios up to 500 have been widely utilized [9,10]. Among these, AgNWs exhibit exceptional electrical conductivity and can be applied in mechanical, optical, thermal, electro-magnetic interference (EMI) shielding, and electrical applications [11,12,13]. Furthermore, AgNW nanocomposites offer potent antibacterial properties, high capability, and excellent UV-blocking ability [14,15], making them promising for wearable fabric applications. Nevertheless, combining AgNWs closely with soft fabrics while maintaining durability for wearable sensors and electric heating applications remains a significant challenge. The conductivity of AgNW-coated fabric gradually declines with time, especially when exposed to rubbing and washing during daily use.

To address this issue, researchers have investigated various methods to enhance the loading and binding fastness of AgNWs onto treated fabrics, including plasma-induced vapor phase graft polymerization of acrylic acid, self-assembled monolayers of phenyl azoles, methoxy silane, and methyl alkane, and the fabrication of multilayered structures based on thermoplastic polyurethane, AgNWs, and reduced graphene oxide [16,17,18]. Moreover, the development of enhanced flexible triboelectric nanogenerators using a multi-layer structure has also been reported [19]. Although many studies have focused on methods for achieving the sustainable stability of smart fabrics by closely connecting conductive materials with a substrate, such as chemical reactions, adhesive reagents, and the assembly of multi-layer structures, a convenient, efficient, and environmentally friendly bonding approach for smart fabrics is still required.

In this study, we utilized AgNWs to fabricate cellulose-based materials via simple hot-pressing enhancement. During hot pressing, the PET fabric reaches its glass transition temperature (T_g_), causing molecular chain movement that results in a sticky flow state. Under high pressure, the AgNWs become embedded into the PET fabric, resulting in a conductive composite fabric with high durability and washability. Our objective is to embed AgNWs into PET fabric by dip coating followed by hot pressing to obtain a smart fabric with excellent conductivity and durability that possesses outstanding electromagnetic shielding, strain sensing, electric heating, and temperature sensing properties, making it a promising candidate for multi-functional smart fabrics. To verify the effectiveness of the hot-pressing process, the resulting non-pressed AgNWs/PET fabrics (NF) and hot-pressed AgNWs/PET fabrics (HF) after washing and adhesive tape experiments are provided.

## 2. Experimental

### 2.1. Materials

Polyvinyl pyrrolidone (PVP, Mw ≈ 58,000, 99%, AR), ethylene glycol (EG, Mw = 98.08, 98%, AR), copper chloride dehydrate (CuCl_2_·2H_2_O, Mw = 170.48, 98%, AR), absolute ethanol (C_2_H_5_OH, Mw = 62.07, 98%, AR), acetone (CH_3_COCH_3_, Mw = 58.08, 99.5%, AR), and silver nitrate (AgNO_3_, Mw = 169.87, 99%, AR) were purchased from Sinopharm Chemical Reagent Co., Ltd., Shanghai, China. Batik professional wax was purchased from Hangzhou Zhiyi Trading Co., Ltd, Hangzhou, China.

### 2.2. Fabrication of AgNWs/PET Fabric

The process of synthesizing AgNWs is illustrated in Figure 1a, and it is the same as in our previous research papers [20,21]. Firstly, the substrate PET fabric is immersed in a solution containing conductive AgNWs, and the process is repeated until the AgNWs/PET fabric achieves good conductivity. Subsequently, the AgNWs/PET fabric is dewaxed using an electric iron on wax-absorbing paper, following which it is subjected to reinforcement by hot pressing.

To isolate the non-conducting part of the periodic structure of the fabric, wax batik printing was employed, which prevented the coating of AgNWs on the targeted areas. This was accomplished by melting the wax using an alcohol lamp and dipping the fabric into the liquid wax, as illustrated in Figure 1a. Once the wax had cooled and solidified onto the PET fabric, it served to block the entry of conductive materials during immersion, effectively creating a periodic structure capable of selective electromagnetic shielding. Three different periodic structures, namely strip grid (SG), (strip patch) SP, and (strips) Ss, were used to achieve this effect. These periodic structures were applied onto the PET fabric substrate using wax batik printing. The hot-pressing process, conducted at both 180 °C and 140 °C for 90 s, led to the formation of the HF, as depicted in Figure 1a. Figure 1b highlights the potential applications of the HF, such as electromagnetic shielding, strain sensing, and electric heating.

### 2.3. Characterizations

The microstructures of the samples were observed using a scanning electron microscope (SEM), model GeminiSEM500, manufactured in Germany. The dynamic resistance of the fabric was measured at room temperature utilizing a resistance meter, model DMM 6500, manufactured by Tektronix Co., Ltd., Shanghai, in China. The washability of the samples, including the HF and NF, was investigated using the Y089D Fully Automatic Washing Shrinkage Tester, manufactured by Wenzhou Fangyuan Instrument Co. Ltd., Wenzhou, China. The electromagnetic interference shielding effectiveness (EMI SE) was evaluated by utilizing the FY800 Fabric EM Shielding Properties Tester, also manufactured by Wenzhou Fangyuan Instrument Co. Ltd., Wenzhou, China. Additionally, the electric heating temperature of the fabrics was measured using an infrared thermal imager (Infra Tec, FLIR System, Dresden, Germany). Overall, these instruments were selected to conduct precise and accurate measurements and observations of the textile samples being examined.

The periodic GAP fabric is a promising material due to its unique properties and potential applications. The synthesis of AgNWs and the wax coating procedure of the AgNWs/PET fabric, as shown in Figure 1a, enable the fabrication of a high-quality, well-controlled periodic GAP fabric. Figure 1b illustrates the versatility of the material with various applications, ranging from EMI shielding to sensing applications such as temperature, joule heating, and strain sensing. These applications are of great importance in many fields of science and engineering, including materials science, electronics, energy storage, and biomedical engineering. Overall, the schematic illustration presented in Figure 1 provides valuable insights into the structure, preparation, and applications of the periodic GAP fabric, highlighting its significant potential for various technological advancements.

## 3. Results and Discussion

### 3.1. Conductivity and Electromagnetic Shielding

Figure 2 displays the surface morphology differences between the HF and NF. The SEM images of the NF (Figure 2a) reveal the attachment of AgNWs to the PET surface. This can be attributed to the adhesive ability between AgNWs and PET, which is due to the Vander Waals force and the residual PVP molecular chain from the preparation of the AgNW solution forming a hydrogen bond with the oxygen in the hydroxyl group (C=O) on the polar group of the PET fiber surface. However, as depicted in Figure 2a,b, AgNWs are prone to fall off when the fabric is subjected to an external friction force, causing the conductivity of the AgNWs to deteriorate over time due to oxidation. Conversely, Figure 2c depicts the surface morphology of the HF under hot pressing at 140 °C for 90 s, where the AgNWs are slightly embedded into the PET fabric. The diagram in Figure 2d further shows that only tiny amounts of the AgNWs are embedded in the PET fabric. However, under the hot-pressing conditions of 180 °C for 90 s, as shown in Figure 2e,f, the AgNWs are mostly covered by PET. At high temperatures, the PET becomes more viscous due to its glass transition temperature (T_g_) being around 80 °C and its ironing temperature being 160 °C. T_g_ represents the relaxation phenomenon of the amorphous part of the polymer from a freezing state to a thawing state, which is a second-order phase transition. As the temperature nears the viscous flow temperature (T_f_) of PET during hot pressing at 180 °C, the surface of the PET fabric becomes partially viscous fluid, and as a result, the AgNWs can be more easily embedded into the fabric under the heating press. Therefore, the degree of embedding of the AgNWs in the fabric under hot pressing at 180 °C in Figure 2f is greater than that under hot pressing at 140 °C, as shown in Figure 2d. Moreover, the conductivity increases with an increase in the hot-pressing temperature. Hence, it is evident that the hot-pressing process under conditions of 180 °C for 90 s can tightly bond the AgNWs with the fabric and enhance the durability and conductivity of the AgNWs/PET fabric.

Figure 2g depicts a schematic diagram of the hot-pressing process, where AgNWs are embedded into the PET fabric in a closely packed manner. The hot-pressing device consists of two copper plates and the power source. The temperature and time can be controlled via the main figure, which heats the copper plates to the desired temperature. In order to investigate the enhancements in conductivity, electromagnetic shielding, and durability of the HF and NF throughout the hot-pressing process, conductivity (Figure 2h) and electromagnetic shielding effectiveness (EMI SE) (Figure 2i,j) measurements are presented. All samples in this section were subjected to hot pressing under 180 °C for 90 s, unless stated otherwise. Figure 2h compares the conductivity and EMI SE of the NF and HF, demonstrating that the conductivity of the AgNW fabric increases from 94.9 S/m to 464.2 S/m after hot pressing. This indicates that the hot-pressing process can enhance the conductivity of the AgNWs/PET fabrics. Additionally, the average EMI SE values of the HF and NF are 25.5 dB and 17.2 dB, respectively, due to the improved electrical conductivity. Therefore, the hot-pressing technology can effectively improve both the conductivity and EMI SE of the AgNWs/PET fabrics.

Furthermore, the wax coating method was utilized to achieve selective SE with three periodic structures, namely SG, SP, and Ss. As illustrated in Figure 2h, the SE peak value of the NF with the SG structure was 27 dB at 1580 MHz, while the corresponding SE peak value of the HF with the SG fabric was 32 dB at 1550 MHz. Similarly, the SE peak value of the NF with the Ss structure was 22.8 dB at 1580 MHz, whereas the SE peak value of the HF with the Ss fabric was 29 dB at 1550 MHz. On the other hand, the SP structure belongs to the series/bandpass type, where the electromagnetic wave is transmitted at a low frequency. Therefore, as depicted in Figure 2j, the average electromagnetic SE of the HF and SP fabrics were 2.2 dB and 1.3 dB, respectively. Additionally, Figure 2j shows that the average SE from 0.3 GHz to 3 GHz for the HF with the SG structure was 25.6 dB, and for the NF with the SG structure, it was 17.2 dB. The average SE of the HF with the Ss structure was 13.2 dB, and for the NF with the Ss structure, it was 9.4 dB. It can be inferred that the value of electromagnetic shielding for the HF was greater than for the NF. Therefore, it can be concluded that the hot-pressing method has the capability to enhance the conductivity and SE of a fabric with AgNWs.

### 3.2. Stick-Resistant Peeling Performance

In wearable fabrics, daily friction is inevitable and can cause wear and tear over time. To simulate this process, adhesive tape was used, as shown in Figure 3. The conductivity variation of the HF and NF with different adhesive tape cycles was measured and is presented in Figure 3b. The tape was pasted on the AgNW fabric with the same pressure applied by a weight and then removed from the fabric. The results indicate that the conductivity of both the HF and NF decreases with increasing adhesive cycles. After five cycles, the conductivity of the HF (at 140 °C) decreased from 440 S/m to 12 S/m, and the conductivity of the HF (at 180 °C) decreased from 464 S/m to 32 S/m. In contrast, the conductivity of the NF decreased to around 1 S/m after being taped five times. The ratio of the conductivity of the HF to the conductivity of the NF under different adhesive cycles is shown in Figure 3c, where the conductivity ratio changes gradually with increasing adhesive cycles. The results suggest that the hot-pressing process enhances the durability of the fabric compared to without hot pressing. Notably, the conductivity of the hot-pressed fabric under 180 °C for 90 s is comparable to that of the fabric hot pressed for 5 min at the same temperature.

### 3.3. Water Washing Fastness

The water fastness of fabrics coated with nano conductive material, specifically AgNWs, was investigated. The washability of the HF and NF wearable smart fabrics is explored in Figure 4 to determine whether hot pressing can enhance the durability of AgNWs/PET fabrics. Figure 4a shows the conductivity variation of the HF and NF under zero, three, six, and nine washing cycles, with the conductivity of the HF being greater than that of the NF. The conductivity of the HF is 464 S/m, and after three, six, and nine washing cycles, the conductivities are 168 S/m, 109 S/m, and 98 S/m, respectively. In contrast, the original conductivity of the NF is 95 S/m, and after washing it six and nine times, the resistance is high and the conductivity is poor. It can be concluded that the conductivity of the HF is superior to that of the NF after washing.

Figure 4b displays the ratio of the conductivity of the HF to that of the NF under different washing cycles. The conductivity ratio of the HF to NF is 4.9 without washing, and after washing for three, six, and nine cycles, the ratio is 21.9, 239, and 279, respectively. The ratio of the conductivity of the HF to NF increases with washing times, indicating that the hot-pressing technique can enhance the stable and durable performance of the AgNW fabric.

Additionally, Figure 4c further investigates the SE of the HF/NF after fabric washing nine times. The results show that the Ss structure provides a maximum of 17.0 dB at 1630 MHz, and the SG structure offers a maximum of 17.3 dB at 1550 MHz. After washing, the wax-coated grid fabric possesses selective electromagnetic shielding properties. However, SP lacks electromagnetic SE performance due to its structure being a capacitor and inductor in series, belonging to the bandpass type, which transmits electromagnetic waves at low frequencies. Figure 4d compares the average electromagnetic SE of the HF with the NF at different washing cycles. The average SE values of the NF and HF are 17.2 dB and 25.6 dB without washing. After three washing times, the average SE of the NF and HF is 0.5 dB and 18.9 dB, respectively. After six washing times, the average SE of the NF and HF is 0.4 dB and 11.7 dB, respectively. After nine washing times, the average SE of the NF and HF is 0.3 dB and 9.5 dB, respectively. It can be concluded that the electromagnetic SE of the HF after washing is greater than that of the NF. Figure 4e demonstrates that the AgNWs remain in the HF fabric after washing. The surface morphology of the HF and NF after nine washings shows a significant number of AgNWs in the HF, while there is a minimal number of AgNWs on the NF. The AgNWs on the NF surface are prone to falling off under the friction of the accompanying cloth, and only a small number of AgNWs exist in the deep fibers of the fabric.

In conclusion, the HF retains its conductivity and electromagnetic SE after washing, while the conductivity and electromagnetic SE of the NF are inferior. The hot-pressing technique can enhance the durable and stable performance of AgNWs in PET fabrics, which can be used in wearable industrial production.

### 3.4. Strain Sensing Performance

Figure 5 illustrates the fabric strain sensing performance after being washed nine times, along with the schematic diagram of the testing machine used for fabric sensing (Figure 5b). The resistance changes of the fabric were recorded in real-time by a computer. Figure 5c displays the resistance changes under different stretching states, where the change rate can be calculated by (R_1_ − R_0_)/R_0_, with R_0_ being the original resistance of the HF and R_1_ being the resistance during stretching or weighting.

Figure 5c–e depict the strain sensing of the NF and HF under bending, pressing, and knee bending. It is observed from Figure 5c that when the fabric is bent, the resistance becomes smaller and remains stable as long as it remains bent. This is because stretching decreases the space between yarns and increases the contact points between them, leading to the formation of conductive networks that cause a decrease in resistance during the stretch process. When the NF returns to its original state, the resistance becomes larger (ΔR/R_0_ stabilizes at −0.17%) but remains smaller than the original resistance (ΔR/R_0_ is 0). The HF, on the other hand, exhibits larger changes in resistance (ΔR/R_0_ of −5.8% to −8.6%) than the NF, owing to its better conductivity after washing. The change in resistance also becomes larger with an increase in the stretching degree, indicating that the sensitivity of the HF fabric is better than that of the NF.

Figure 5d,e show the strain sensing abilities of the HF and NF to the compress of copper ingot and the knee bending, with both results exhibiting good reproducibility. The extent of the change in the HF is more obvious than that of the NF, owing to its better conductivity. These results indicate that the sensing effect of the HF is better, making it more suitable for possible use in sensing carpets. According to the weight changes in the carpet per unit area, the sensing carpet can be applied as an automatic alarm to send out an alert for elderly individuals who live alone if they fall, as reported in prior works [22]. Indeed, sensitivity and specificity, the integration of conductive fabric with sensor technologies, data processing and algorithms for motion and fall detection, as well as practical challenges such as power supply, real-time monitoring, and user comfort, are pivotal concerns that demand thorough consideration for the practical realization of the technology. These factors collectively contribute to the feasibility and effectiveness of the technology in the context of human motion sensing and fall detection. Recent research has introduced valuable insights into the potential practical applications of smart fabric technology, including active-matrix flexible fabrics, digital electrochemistry integration, and the architecture of organic transistors in flexible electronics [23,24].

### 3.5. Electrical Heating and Temperature Sensing Performances

Figure 6 illustrates the electrothermal behavior of the HF after undergoing nine cycles of fabric washing. The measurement of temperature sensing and joule heating experimental setup is depicted in Figure 6a, where an electric current is applied to the fabric and the temperature variation is monitored using an infrared camera. Figure 6b presents the real-time temperature change of the NH under a specific current. At a current of 0.02 A, the temperature of the NF reaches 91 °C, while at 0.04 A, it rises to 145 °C. Upon reaching 0.06 A, the fabric ignites and the NH attains its maximum temperature. Figure 6c depicts the temperature variation of the HF under different applied currents, where the stable temperatures of the HF are 50 °C, 68 °C, 90 °C, and 110 °C at currents of 0.02 A, 0.04 A, 0.06 A, and 0.08 A, respectively. Compared to the NF, the temperature stability of the HF is superior, indicating its potential for various applications in the field. The stability of the HF’s temperature is attributed to its superior conductivity compared to the NF. Based on Joule’s equation, Q = I^2^Rt (where Q is heat, I is current, R is resistance, and t is time), when the current is kept constant, smaller R values lead to smaller Q values.

In Figure 6d, the real-time changes in relative resistance, ΔR/R_0_, and temperature are shown when the HF is subjected to a current of 0.08 A. The ΔR/R_0_ of the fabric decreases to 6% with an increase in temperature. The trend in the resistance change is opposite to that of temperature, and the changes in ΔR/R_0_ and temperature correspond to each other. Thus, the HF can be employed as a wearable resistive temperature fabric sensor. To further study the potential applications of the HF, Figure 6e presents the temperature change of the HF caused by wind blowing under a 0.04 A current. After powering on and following an electric heating period, the small fan is activated, directing airflow towards the fabric over the dotted red lines that mark areas. “OFF” refers to the electric heating being off. The HF is placed on a doll, and the temperature rises to 61 °C with the current on. However, the temperature drops to 36 °C with the wind blowing outside, highlighting the potential use of the HF in perceiving external environmental changes in smart fabrics. Furthermore, as depicted in Figure 6f, when the thermochromic ink is coated on the HF, it changes color from red to yellow upon the application of electricity, signifying its potential in temperature sensing applications, such as electric kettles and other similar devices. In summary, the AgNW HF possesses excellent electrothermal properties, making it a promising temperature sensor for wearable smart fabrics.

## 4. Conclusions

In conclusion, this study presents a novel method to improve the durability and stability of AgNW-coated PET fabric and demonstrates its multifunctional properties as a wearable smart fabric. The hot-pressing process greatly enhances the EMI shielding effectiveness, strain sensing capability, and electrothermal properties of the fabric. Specifically, the conductivity of the hot-pressed fabric (at 180 °C for 90 s) reaches 464.2 S/m, surpassing the conductivity of non-hot-pressed fabric, which is measured at 94.9 S/m. Subsequent to undergoing nine washing cycles, the fabric with the specified hot-pressing treatment exhibits notable characteristics. The HF (Ss structure) demonstrates an average SE of 9.5 dB, a conductivity of 98 S/m, and a maximum strain sensing capability of −11.5%. It is important to note that hot pressing significantly contributes to the durability of the AgNWs/PET fabric. Conversely, the NF does not exhibit effective EMI shielding capabilities, underscoring the importance of the hot-pressing process in enhancing the fabric’s multifunctionality and durability. Additionally, the HF is capable of electrical heating up to 110 °C at 0.08 A and can be used in various applications, such as strain sensing for human motion detection. Furthermore, the fabric can be used in wearable temperature sensors based on resistance changes. This HF fabric can be considered a promising candidate for the development of multifunctional wearable smart fabrics, and the hot-pressing process has great potential for industrialization.

## Figures and Tables

**Figure 1 polymers-15-04258-f001:**
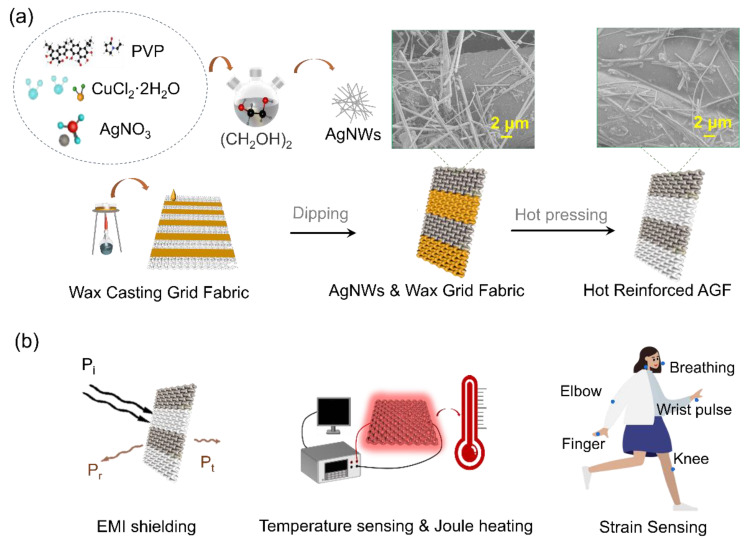
Illustration of the structure, preparation, and application of the periodic AgNWs/PET fabric. Figure (**a**) depicts the fabrication process of AgNWs and the wax coating procedure used for the AgNWs/PET fabric. The SEM images in suspension depict the morphology of the NF (**left**) and HF (**right**), respectively. Figure (**b**) shows the various applications of the fabric, including electromagnetic interference (EMI) shielding, temperature sensing, joule heating, and strain sensing.

**Figure 2 polymers-15-04258-f002:**
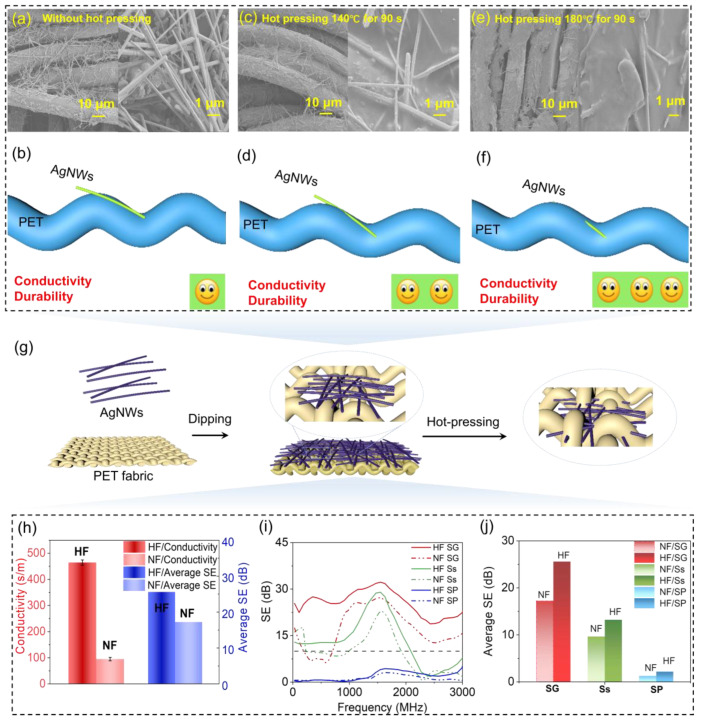
Display of the improvement in the electrical conductivity and electromagnetic SE of the AgNWs/PET fabrics using hot-pressing technology and the wax coating method, along with a schematic representation of the adhesive structure. Specifically, (**a**) showcases SEM images of the NF sample. (**c**,**e**) depict SEM images of the HF samples subjected to hot pressing at 140 °C for 90 s and 180 °C for 90 s, respectively. (**b**,**d**,**f**) present schematic diagrams of the NF sample, the HF sample under hot pressing at 140 °C for 90 s, and the HF sample under hot pressing at 180 °C for 90 s, respectively. (**g**) illustrates the schematic diagram of the hot-pressing process for embedding AgNWs into PET fabric. (**h**) shows a comparison of the conductivity and average SE of HF and NF. (**i**) displays the SE of NF and HF for the periodic structures of SG, SP, and Ss. (**j**) shows the average electromagnetic SE of NF and HF for the SG, SP, and Ss structures.

**Figure 3 polymers-15-04258-f003:**
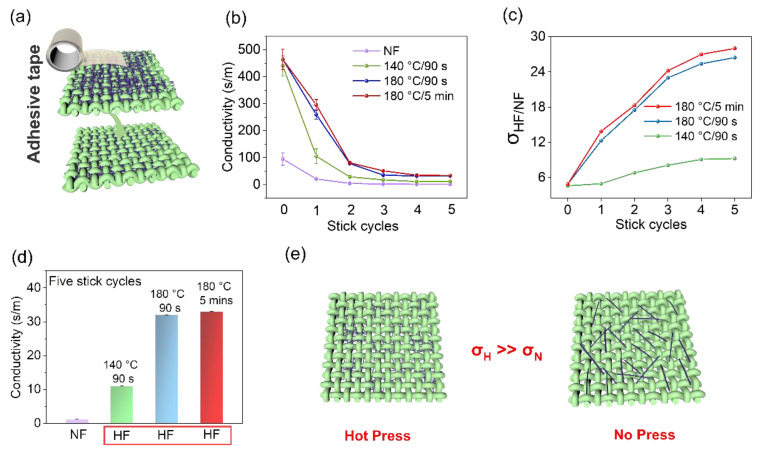
Depiction of the results of the adhesive tape durability test on the conductive properties of AgNWs fabrics. (**a**) illustrates the schematic diagram of the adhesive tape and the fabric. (**b**) presents the conductivity values of the HF and the NF under different stick cycles of adhesive tape. (**c**) displays the conductivity ratio of HF fabric to that of NF under different adhesive cycles. (**d**) shows the conductivity of the four fabrics after five adhesive cycles. Finally, (**e**) presents a diagram of the residual conductive materials on the surface of the fabric after the adhesive cycles.

**Figure 4 polymers-15-04258-f004:**
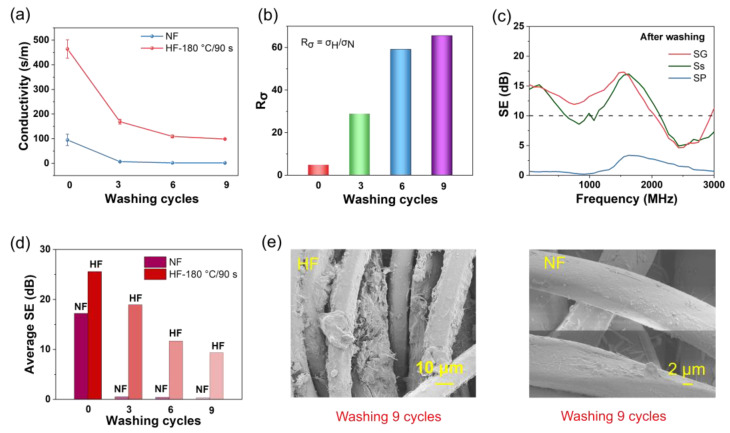
Presentation of the experimental results related to the conductivity and electromagnetic SE of the NF and HF samples under washing cycles. Specifically, (**a**) illustrates the variation in conductivity of the NF and HF after 0, 3, 6, and 9 washing cycles. (**b**) shows the ratio of HF conductivity to NF conductivity under different washing cycles. Furthermore, (**c**) presents the electromagnetic SE of HF for different hybrid fabrics, including SG, SP, and Ss, after 9 washing cycles. (**d**) provides a comparison of the average electromagnetic SE of HF and NF at 0, 3, 6, and 9 washing cycles. Finally, (**e**) displays the scanning electron microscopy (SEM) images of HF and NF samples after undergoing 9 washing cycles.

**Figure 5 polymers-15-04258-f005:**
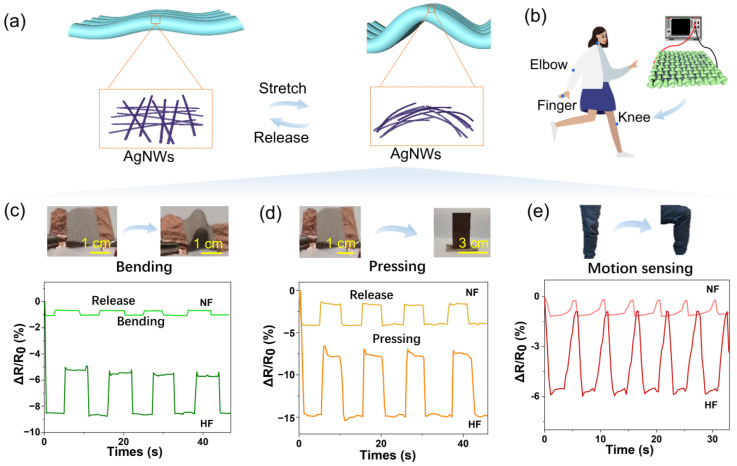
Illustration of the performance of AgNWs/PET fabric strain sensing after repeated washing cycles, along with the schematics of the test setup. (**a**) presents the mechanism of bending-induced stretch and compression modes of the fabric. (**b**) provides a schematic diagram of the electromechanical test system. (**c**,**d**) demonstrate the relative resistance changes of NF and HF under different bending degrees and pressures, respectively. (**e**) shows the strain sensor for knee bending using NF and HF.

**Figure 6 polymers-15-04258-f006:**
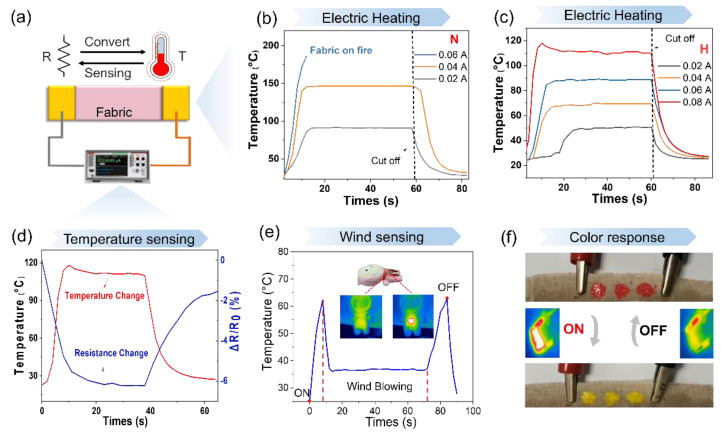
Illustration of the temperature change of two types of film, namely NF and HF, at varying currents. (**a**) displays a schematic diagram depicting the measurement of temperature sensing and joule heating. Specifically, (**b**,**c**) show the temperature change of NF and HF, respectively, at different levels of current. Furthermore, (**d**) presents the real-time changes in relative resistance (ΔR/R_0_) and the temperature change of HF under a 0.08 A current. Additionally, (**e**) shows the temperature change in HF caused by wind blowing under a 0.04 A current. The temperature changes are depicted in (**f**) for a given 0.02 A and 0.08 A current, respectively.

## Data Availability

The data presented in this study are available on request from the corresponding author.

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
