# Peer review of "Multifunctional Silver Nanowire Fabric Reinforced by Hot Pressing for Electromagnetic Interference Shielding, Electric Heating, and Sensing"

_polymers, 2023, doi:10.3390/polym15214258_

Round 1

Reviewer 1 Report

Comments and Suggestions for Authors

In this manuscript, Zeng et al. introduce a novel method to enhance the durability and stability of AgNWs-coated PET fabric. The reported process involves a hot-pressing methodology, which provides the PET fabric with embedded AgNWs. This technology significantly improves the fabric's EMI shielding effectiveness, strain-sensing capability, and electrothermal properties. My recommendation is a moderate revision to address the following comments.

1) After undergoing 9 washing cycles, the fabric structure showed an average SE of 9.5 dB, a 98 S/m conductivity, and a maximum strain sensing capability of -11.5%. Is there any particular reason for performing 9 cycles only? What is the performance at increasing washing cycles (i.e., from 10 to 100 cycles)?

2) The authors have argued that the fabric can be electrically heated up, making it suitable for various applications such as human motion sensing and fall detection, besides being a suitable choice for the development of multifunctional wearable smart fabrics. Considering these application scenarios, how far the reported technology would be from the practical realization of human motion/fall detection? Please append such important considerations in the revised manuscript.

3) Still, on the previous topic, some recent papers (doi.org/10.1002/adma.202101272 and doi.org/10.1002/adma.202204804) may add insightful pictures for possible practical applications of the hot-pressing AgNW-PET fabric technology (e.g. active-matrix flexible fabrics). Particularly, the former is an article about digital-electrochemistry integration, whereas the latter is a review paper about the organic transistor-architecture roles in flexible electronics. The incorporation of these aspects in the discussion will certainly improve the significance of this manuscript as well as the interest of the readers.

4) Minor points:

(a) "Nano silver wire fabric" or "Silver nano-wire (AgNW) fabric"?

(b) Is the aspect ratio given by length/diameter (p.2, l.50)? Please specify it.

(c) SEM images in Fig.1a must be detailed in the caption. Please improve their resolutions and provide scale bars.

(d) Please define all acronyms (such as "EMI" in p.2, l.52) at the moment they first appear throughout the main text.

(e) In Fig. 5, please provide scale bars for the photos.

(f) Please check the figure labeling to avoid misleading references (e.g. "Fig.5(f,g,h)" is not labeled correctly).

Comments on the Quality of English Language

A minor editing of the English is required.

Reviewer 2 Report

Comments and Suggestions for Authors

We received paper from Polymers (polymers-2683321) with the title “Multifunctional Nano-silver Fabric Reinforced by Hot Pressing for EMI Shielding, Electric Heating and Sensing”. The paper was interesting and have novelty on the content. However, after carefully check the manuscript, several aspects need to be revised:

11.       In the abstract, it is said that the process was used hot-pressed fabric. What the temperature so that the authors identified as hot-pressed?

22.       It is shoed that excellent negative strain performance (15%), how about in term of positive strain performance? How was the results?

33.       The summarized related to the Ag nano particles need to be improve related to the application of the Ag and how the manufacturing process. Several methods are introduced and give high purity. The following work can be add as a reference: Performance evaluation of Ag/SnO2 nanocomposite materials as coating material with high capability on antibacterial activity

44.       In the materials, is the silver nitrate have high purity? How much is it?

55.       How about other materials as well? Please stated.

66.       In the method, what temperature was used in the process?

77.       In the results, it is showed that by increase the temperature, and time, the conductivity can be increase. Why was the study limited the hot press temperature with 180? Why not 200 or 220?

88.       How the wind blowing testing was applied?

99.       In the conclusion, stated all the important results and placed in a single paragraph or in dots mode.

Round 2

Reviewer 1 Report

Comments and Suggestions for Authors

The authors provided a point-by-point answer to all my queries. It's worth pointing out that your answer to my first question is particularly inconclusive. However, this specific question is not crucial to reach an adequate discussion of the results and conclusions, but rather a curiosity for the Reviewer. In summary, the manuscript was substantially improved during the previous round of review. My recommendation is accepted in its current form.

Comments on the Quality of English Language

Minor editing of the English is required.